# Whole genome sequencing characterization of Slovenian carbapenem-resistant *Klebsiella pneumoniae,* including OXA-48 and NDM-1 producing outbreak isolates

Katarina Benulič[1]*, Mateja Pirš[1], Natacha Couto[2], Monika Chlebowicz[2], John W. A. Rossen[2], Tomaž Mark Zorec[1], Katja Seme[1], Mario Poljak[1], Tatjana Lejko Zupanc[3], Eva Ružić-Sabljić[1], Tjaša Cerar[1]

1 Faculty of Medicine, Institute of Microbiology and Immunology, University of Ljubljana, Ljubljana, Slovenia, 2 Department of Medical Microbiology and Infection Prevention, University Medical Center Groningen, University of Groningen, Groningen, Netherlands, 3 Department of Infectious Diseases, University Medical Center Ljubljana, Ljubljana, Slovenia

* katarina.ben@gmail.com

**Data Availability Statement:** The next-generation sequencing data are available from European Nucleotide Archive (study accession number

## Abstract

### Objectives

The first hospital outbreak of carbapenemase-producing Enterobacteriaceae in Slovenia occurred in 2014–2016. Whole genome sequencing was used to analyse the population of carbapenem-resistant *Klebsiella pneumoniae* collected in Slovenia in 2014–2017, including OXA-48 and/or NDM-1 producing strains from the outbreak.

### Methods

A total of 32 *K. pneumoniae* isolates were analysed using short-read sequencing. Multi-locus sequence typing and core genome multi-locus sequence typing were used to infer genetic relatedness. Antimicrobial resistance markers, virulence factors, plasmid content and *wzi* types were determined. Long-read sequencing was used for six isolates for detailed analysis of plasmids and their possible transmission.

### Results

Overall, we detected 10 different sequence types (STs), the most common being ST437 (40.6%). Isolates from the initial outbreak belonged to ST437 (12/16) and ST147 (4/16). A second outbreak of four ST15 isolates was discovered. A new ST (ST3390) and two new *wzi* types (*wzi*-556, *wzi*-559) were identified. *bla*OXA-48 was found in 17 (53.1%) isolates, *bla*NDM-1 in five (15.6%), and a combination of *bla*OXA-48/NDM-1 in seven (21.9%) isolates. Identical plasmids carrying *bla*OXA-48 were found in outbreak isolates sequenced with long-read sequencing technology.

PRJEB32863). The rest of the relevant data are within the paper and its Supporting Information files.

**Funding:** This work was supported by the Slovenian Research Agency, grant number P3–0083 (KB, MP, TMZ, KS, MP, ERS, TC) - www.arrs.si The funders had no role in study design, data collection and analysis, decision to publish, or preparation of the manuscript.

**Competing interests:** The authors have declared that no competing interests exist.

## Conclusions

Whole genome sequencing of Slovenian carbapenem-resistant *K. pneumoniae* isolates revealed multiple clusters of STs, two of which were involved in the first hospital outbreak of carbapenem producing *K. pneumoniae* in Slovenia. Transmission of the plasmid carrying *bla*OXA-48 between two STs was likely to have occurred. A previously unidentified second outbreak was also discovered, highlighting the importance of whole genome sequencing in detection and/or characterization of hospital outbreaks and surveillance of drug-resistant bacterial clones.

## Introduction

Antimicrobial resistance is a matter of concern worldwide and carbapenemase-producing Enterobacteriaceae (CPE) pose a major threat to human health. Resistance to carbapenems can be caused by production of carbapenemases (KPC, NDM, OXA-48, VIM), spreading mainly due to acquired plasmids or other mobile elements, and permeability alterations caused by the loss of porins and overexpressed efflux system [1,2].

In recent years, increased resistance to carbapenems in *Klebsiella pneumoniae* has been reported in several European countries, with variable distribution of predominant types of car-bapenemases [1,3,4]. The most common *K. pneumoniae* clonal groups (CGs), as defined by multi-locus sequence typing (MLST), which are associated with outbreaks, are CG258 (sequence type (ST) 258 and its derivatives, including ST11, ST340 and ST437), CG14/15, CG17/20, CG43 (including ST101) and ST147 [5]. A recent study shows that the most common clonal lineages in Europe are indeed ST11, ST15, ST101, ST258 and their derivatives [4], and also an emerging high-risk clone ST307 [6,7]. High-risk clones ST258, ST14, ST37, ST147 and ST101 are associated with carbapenemase resistance, while ST15 and ST17 usually carry extended spectrum beta-lactamases [8].

In Slovenia, a small Central European country with 2 million inhabitants, systematic laboratory surveillance of carbapenem-resistant (CR) Enterobacteriaceae began in the second half of 2010. Until late 2014, only sporadic cases of CPE were detected (up to 10 patients with CPE per year), isolated mainly from surveillance samples. Colonization and/or infection with such strains was most frequently associated with previous hospitalization abroad, notably Serbia [9]. Slovenia lies at the eastern border of Italy and northern border of the Balkans, where the epidemiological situation has been worsening for years, with most countries reporting at least sporadic hospital outbreaks and two countries reporting interregional spread [10,11]. The worst affected country in the Balkans is Serbia with high incidence of CR-*K. pneumoniae* [10], mainly NDM-1 and OXA-48 producers [3,12]. Italy is also an endemic country with predominately KPC carbapenemases [3,4,11], which has recently also experienced a significant NDM outbreak [13].

In Slovenia, as elsewhere, *K. pneumoniae* was the most frequently isolated CPE species (50%), followed by *Enterobacter* spp. (25%) and *Escherichia coli* (17%) [6,9,14]. During the period 2014–2017, a total of 91 patients with CP-*K. pneumoniae* were identified, almost half of those were part of the first Slovenian hospital outbreak of CPE which began at the end of October 2014 in the largest tertiary teaching hospital, lasting until February 2016. A total of 40 patients were affected: OXA-48- and/or NDM-producing *K. pneumoniae* were isolated from 31 patients, CP-*K. pneumoniae* and CP-*E. coli* producing OXA-48 and/or NDM-1 were

simultaneously present in a further seven patients, and OXA48-producing *E. coli* alone was detected in two. Two patients had also other CPE in combination with CP-*K. pneumoniae*. The outbreak was investigated with classic epidemiological investigation and genotyping of the isolates was performed using pulse-field gel electrophoresis and MLST determination [15].

While data obtained using classical approach can provide a general overview of the situation, data obtained using whole genome sequencing (WGS) gives much more detailed and relatively rapid insight into the situation as WGS not only has much higher discriminatory power than PFGE [16], but also allows simultaneous detection of antimicrobial resistance genes, plasmids and virulence factors. We have thus used WGS to assess the population structure of CR-*K. pneumoniae* between 2014 and 2017, and to further elucidate the first hospital CPE outbreak with OXA-48 and NDM-1 carbapenemases. We analysed the genomes of selected *K. pneumoniae* isolates to determine their relatedness and to detect possible high-risk clones, antimicrobial resistance markers, virulence factors and plasmid content. We used long-read sequencing on selected outbreak isolates to look for horizontal spread of a transmissible plasmid between these isolates.

## Materials and methods

### Selection of isolates

A total of 32 *K. pneumoniae* isolates were included in this study. Outbreak isolates were selected based on PFGE profiles [15]. The remaining isolates were selected to reflect the Slovenian population of carbapenemases. All were chosen from the laboratory collection of Institute of Microbiology and Immunology in Ljubljana, which serves as the Slovenian national expert laboratory, between 2014 and 2017 (S1 Table). Our selection included 16 CP-*K. pneumoniae* isolates obtained during the hospital outbreak and 16 CP-*K. pneumoniae* isolates unrelated to the outbreak. Six isolates were selected for long-read sequencing, five for the analysis of the *bla*OXA-48 plasmid and one for the confirmation of the *bla*LEN gene (S1 File).

### Routine bacterial identification and antimicrobial susceptibility testing

MALDI-TOF mass spectrometry (Microflex LT with regularly updated Brucker MS library Brucker Daltonics, Bremen, Germany) was used for identification of organisms. Disk diffusion was used for antimicrobial susceptibility testing according to contemporary EUCAST guidelines (www.eucast.org), 2014–2017, with the ending of the isolate name denoting the year of isolation.

### Molecular detection of blaNDM, blaKPC, blaOXA-48-like, blaIMP and blaVIM

Multiplex real-time PCR targeting the genes *bla*NDM, *bla*KPC, *bla*OXA-48-like, *bla*IMP and *bla*VIM was performed using the LightMix Modular Carbapenemase kits (TIB Molbiol, Berlin, Germany) [15].

### Short-read whole-genome sequencing

QIAamp DNA Mini Kit (Qiagen, Hilden, Germany) was used to extract genomic DNA for short-read WGS. Fourteen libraries were prepared using the Nextera XT DNA Library Prep Kit (Illumina, San Diego, CA, USA). A further 18 libraries were prepared using the Nextera DNA Flex Library Prep Kit (Illumina, San Diego, CA, USA). Sequencing was performed on the Illumina MiSeq Platform (2 x 300 bp). FastQC 0.11.8 (https://www.bioinformatics.babraham.ac.uk/projects/fastqc/) was used for quality control of raw reads. Default settings for

trimming were used in Ridom SeqSphere 5.1.0. (Ridom, GmbH, Munster, Germany). Briefly, trimming was performed on both ends of the reads until the average base quality was > 30 in a window of 20 bases. Reads were subsequently assembled *de novo* using the Velvet assembly tool in Ridom SeqSphere 5.1.0. (Ridom GmbH, Münster, Germany) with default settings [17]. Assembly quality check was performed by Ridom SeqSphere and Quast programme [18] (S1 Table).

### Long-read whole genome sequencing using Oxford tanopore technologies

DNA for long-read sequencing by Oxford Nanopore Technologies (ONT, Oxford, UK) was isolated using the PureLink Genomic DNA Mini Kit (Invitrogen, Thermo Fisher Scientific, Carlsbad, CA, USA), following the manufacturer's instructions. Libraries were prepared using Native Barcoding (EXP-NBD104) and Ligation Sequencing Kits (SQK-LSK109) (ONT). DNA products were sequenced in a GridION X5 system (ONT) on a FLO-MIN106 flow cell for 48 h. Basecalling was conducted using Guppy v2.0.5 (ONT), and Porechop v0.2.3_seqan2.1.1 was used for sequence trimming and to demultiplex the dataset (https://github.com/rrwick/Porechop). Original Illumina reads were trimmed with Trimomatic v0.39 [19]. Hybrid *de novo* sequence assemblies were obtained using Unicycler v0.4.7 [20].

### MLST genotyping, wzi typing, virulence genes, resistance genes and plasmid detection

MLST [21], *wzi* typing and detection of genes encoding virulence factors were achieved by uploading contigs to the Pasteur Institute website (http://bigsdb.pasteur.fr) [22–24]. Virulence score was calculated according to Kleborate software (https://github.com/katholt/Kleborate). Determination of resistance genes, detection of plasmids and pMLST were performed with default settings using ResFinder 3.1 [25], PlasmidFinder 2.0 and pMLST 1.4 (https://cge.cbs.dtu.dk/services/) [26], respectively. New ST and *wzi* types were submitted to the Pasteur Institute website, where new identification numbers were assigned. Six isolates sequenced using ONT were also analysed using ResFinder 3.1 [25], PlasmidFinder 2.0 and pMLST 2.0 [26]. The Bandage programme was used for identification of carbapenemase genes [27]. Plasmids of five isolates positive for *bla*OXA-48, which were sequenced using ONT, and a reference plasmid from the Kp11978 strain [28,29], were compared and visualized using the BLAST Ring Image Generator (BRIG) [30]. The plasmid was annotated using Geneious 8.1.8 (Biomatters, Ltd., Auckland, New Zealand) and annotations were additionally compared to non-redundant protein sequences database using blastx at NCBI (http://blast.ncbi.nlm.nih.gov). Our five aforementioned isolates were also compared in EasyFig [31].

### Nucleotide sequence accession number

Sequencing data have been submitted to the European Nucleotide Archive (study accession number PRJEB32863).

## Results

### Real-time PCR to detect AMR genes

According to real-time PCR, the genes *bla*OXA-48-like and *bla*NDM-1 were the most commonly determined, found in 17/32 (53.1%) and 5/32 (15.6%) isolates, respectively (Table 1). Seven out of thirty-two (21.9%) isolates carried both genes. *bla*KPC-2 and *bla*VIM-1 genes were found in one isolate each.

**Table 1. WGS characterization of *K. pneumoniae* isolates (n = 32), carbapenem-resistance and *wzi* typing.**

| Isolate | ST | *wzi* typing | *bla*KPC-2 | *bla*VIM -1 | *bla*NDM-1 | *bla*OXA-48 | *bla*OXA-181 |
|---|---|---|---|---|---|---|---|
| BR318-14 | 15 | 24 | | | + | | |
| BM670-16 | 15 | 24 | | | + | | |
| BM367-17 | 15 | 24 | | | + | | |
| BR402-14 | 15 | 447 | | | + | | |
| BM433-16 | 35 | 37 | | | | | + |
| BR406-15 | 37 | 83 | | | | + | |
| BR4-14 | 101 | 137 | | | + | + | |
| BR605-15 | 101 | 137 | | | | + | |
| BR615-16 | 101 | 137 | | | | + | |
| BM679-17 | 101 | 137 | | | + | | |
| BR370-14[a] | 147 | 64 | | | | + | |
| BR387-14[a] | 147 | 64 | | | | + | |
| BR328-14[a] | 147 | 64 | | | | + | |
| BR319-14[a] | 147 | 64 | | | | + | |
| BR193-17 | 147 | 99 | | | | + | |
| BM230-17 | 258 | 29 | + | | | | |
| BR470-15 | 268 | 95 | | | | + | |
| BR321-14[a] | 437 | 109 | | | | + | |
| BR329-14[a] | 437 | 109 | | | | + | |
| BR38-15[a] | 437 | 109 | | | | + | |
| BR103-15[a] | 437 | 109 | | | | + | |
| BR179-15[a] | 437 | 109 | | | + | + | |
| BR194-15[a] | 437 | 109 | | | + | + | |
| BR211-15[a] | 437 | 109 | | | + | + | |
| BR247-15[a] | 437 | 109 | | | | + | |
| BR252-15[b] | 437 | 109 | | | + | + | |
| BR254-15[a] | 437 | 109 | | | + | + | |
| BR76-16[a] | 437 | 109 | | | | + | |
| BR282-16[a] | 437 | 109 | | | | + | |
| BR207-15[a] | 437 | 109 | | | + | + | |
| BR380-15 | 2384 | 559 | | + | | | |
| BR737-16 | 3390 | 556 | | | | + | |

[a] outbreak isolates

[b] additionally determined outbreak isolate

## Detection of beta-lactamase genes in WGS data

Overall, we detected 30 different beta-lactam resistance genes using WGS (Table 1, S2 Table). The majority of beta-lactamase genes were rare, with 16 genes being present in one isolate each. The most commonly detected was *bla*CTX-M-15, which was found in 27/32 (84.4%) isolates. In one *K. pneumoniae* isolate, *bla*LEN gene was detected. Carbapenemase genes detected with WGS were concordant with the results of the real-time PCR.

## Multi-locus sequence typing

Based on short-read WGS data, we detected 10 different STs (Table 1, Fig 1). The most frequent type was ST437 (13/32; 40.6%). Outbreak isolates belonged to ST437 (12/16) and ST147

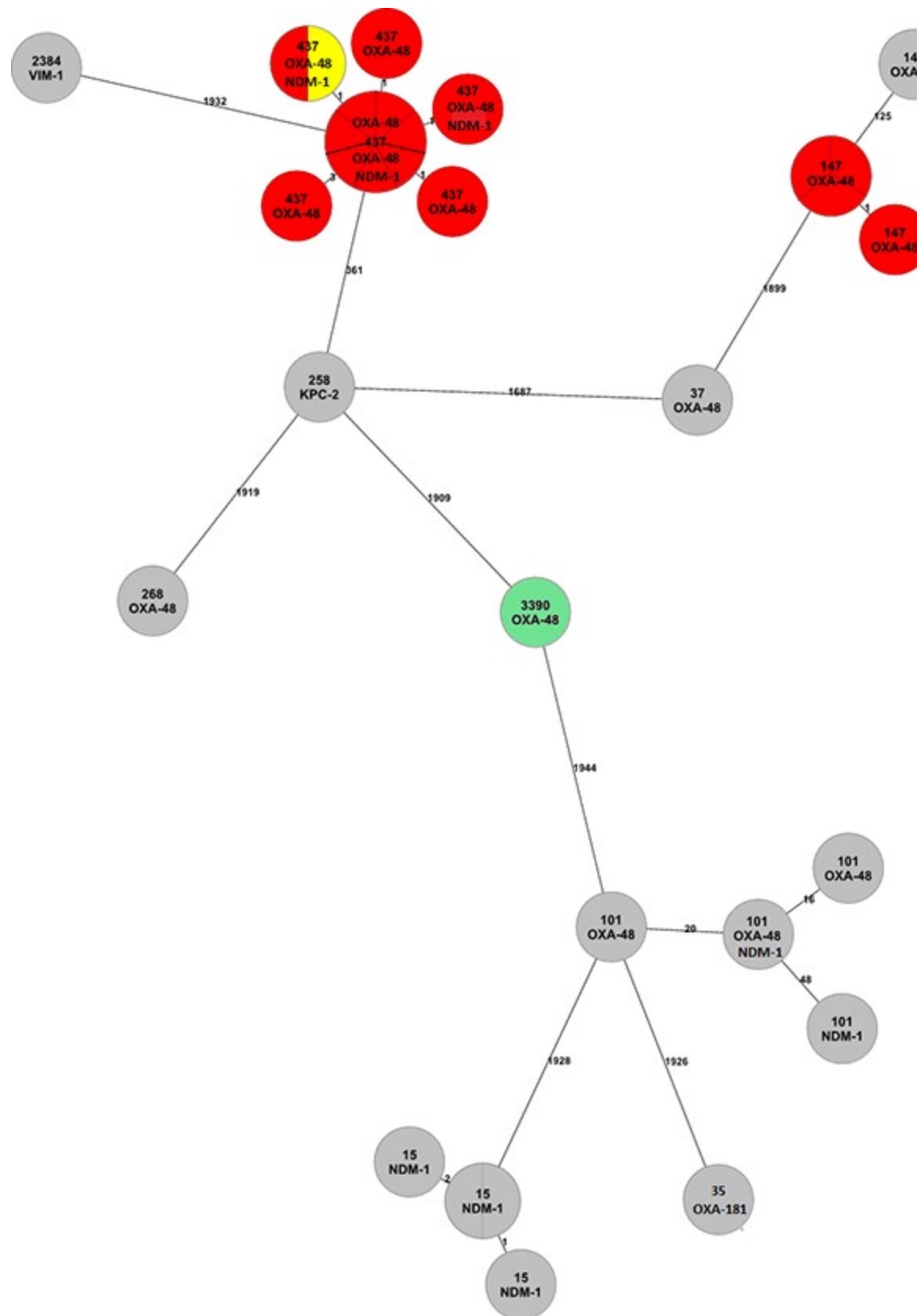

**Fig 1. cgMLST minimum spanning tree of 32 Slovenian *K. pneumoniae* isolates, calculated in Ridom SeqSphere and based on comparing 2365 alleles, ignoring pairwise missing values.** Each node is labelled with an ST and carbapenemase gene. Grey nodes represent isolates not related to the outbreak. Red nodes are epidemiologically confirmed outbreak isolates. The yellow node is an additionally determined outbreak isolate by WGS. Green node represents newly determined ST. Lines indicate the number of allelic differences.

(4/16). One of the ST437 isolates (marked in yellow, Fig 1) was initially considered as unrelated to the outbreak according to the epidemiological data, though it clustered together with other ST437 isolates from the outbreak. The four ST147 outbreak isolates did not cluster with the rest of the outbreak isolates and the fifth ST147 isolate was not related to the outbreak. Interestingly, a second outbreak of four closely related isolates of ST15 was discovered. Further four isolates belonged to ST101 but were not closely related to each other. The remaining isolates (6/32) were assigned to different STs, one of which was new, namely ST3390 (*gapA*:2; *infB*:1; *mdh*:1; *pgi*:1; *phoE*:361; *rpoB*:1; *tonB*:14).

### Virulence factors

Complete *mrk* clusters were detected in 24/32 (75.0%) isolates (S2 Table). All 32 isolates carried *mrkB*, *mrkC*, *mrkF* and *mrkJ*. Complete *ybt* (yersiniabactin) cluster was detected in 5 isolates, for which a virulence score 1 was calculated. Complete *iuc* (aerobactin) locus was detected in four isolates, three of them belonging to the outbreak. Colibactin and salmochelin coding loci were not detected among our isolates. In 27/32 (84.4%) isolates, including all outbreak isolates, we calculated a virulence score 0. Interestingly, ST437 and ST147 outbreak isolates possessed a maximum of 11 virulence genes, with the majority carrying eight genes.

### Wzi typing

Overall, 12 different *wzi* types were identified, two of them being new variants, namely *wzi*-556 and *wzi*-559. The most frequently detected was *wzi*-109, which was assigned to all (13/13) ST437 isolates (Table 1). Four of five closely related ST147 isolates shared an identical *wzi*-64 type; the fifth isolate was *wzi*-99.

### Plasmid content

Plasmid analysis revealed a high diversity of incompatibility (Inc) groups (n = 23) (S2 Table). The most frequent was IncL/M (pOXA-48), detected in 23/32 (71.9%) isolates. IncL/M (pOXA-48) was not detected in two of 24 *bla*OXA-48-positive isolates. All but one (11/12) NDM-1-positive isolates had detectable IncA/C$_2$, although other plasmid groups were also detected, e.g. IncFII(K), IncFIB(K), IncHI1B (S2 Table).

Plasmid analysis of five outbreak isolates sequenced with ONT revealed an almost identical plasmid in the Kp11978 reference and all five isolates (ST437, ST147) with a *bla*OXA-48 gene detected (Fig 2). The *bla*OXA-48 gene, along with the gene for acetyl CoA carboxylase and transcription regulator *lysR* was embedded between the two IS*1999*. The only difference between our isolates was present in isolate BR38, which had a *mucAB* region inserted (S1 Fig). In one of the aforementioned five ONT sequenced isolates, *bla*NDM-1 carrying plasmid was also detected besides *bla*OXA-48 plasmid. In the sixth isolate neither of the two plasmids was detected, nor the *bla*LEN gene, which was detected with Illumina sequencing.

### Discussion

The most represented ST in CP-*K. pneumoniae* from this study was ST437, a single locus variant of ST258 from the widely distributed clonal group CG258 [32]. This is due to the first

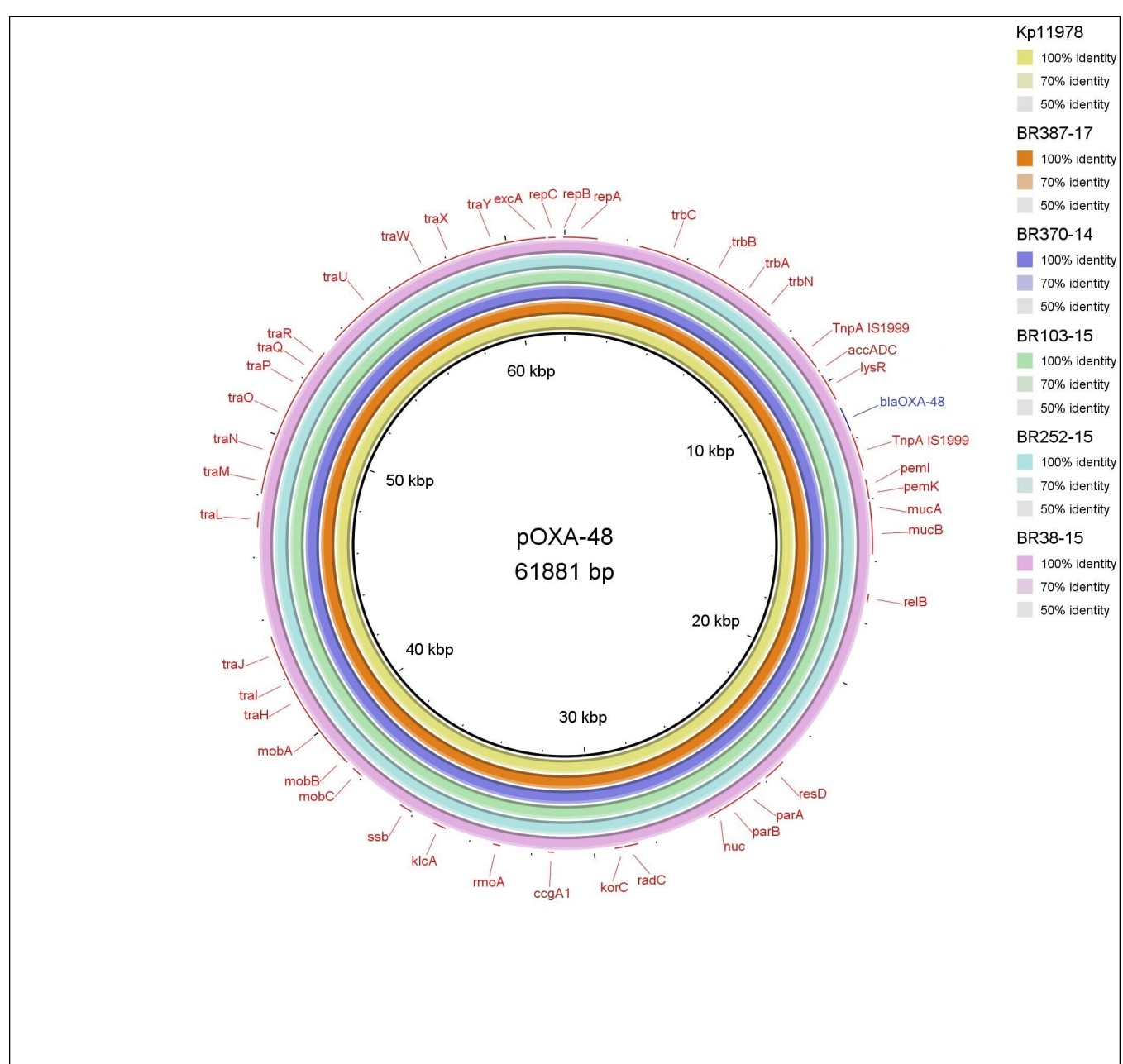

**Fig 2. BRIG-generated schematic representation of the plasmid encoding *bla*OXA-48 (marked in blue), sequenced using hybrid assemblies of five *Klebsiella pneumoniae* isolates and a reference plasmid from Kp11978 isolate [28,29].**

hospital outbreak with aforementioned ST437 and ST147, that occurred in 2014–2016 [15], with OXA-48- and/or NDM-1-producing *K. pneumoniae*. Almost identical plasmid was confirmed in different outbreak isolates belonging to the two STs suggesting plasmid-mediated spread of *bla*OXA-48 resistance genes. This is in concordance with initial molecular analysis of the outbreak where PFGE demonstrated oligoclonal structure in epidemiologically clearly linked patients [15]. A recent study shows that the most common and high-risk clonal lineages in Europe are ST11, ST15, ST101 and ST258 and their derivatives [4], of which STs 15, 101 and 258 were also detected among our isolates. ST307, the cause of the recent outbreak with

OXA-48/NDM-1 producing and colistin-resistant *K. pneumoniae* in Germany [7], was not found among our isolates.

WGS analysis revealed an additional outbreak isolate (BR252-15). Epidemiological investigation did not establish any connection between the patient, from whom this *K. pneumoniae* ST437 was isolated, and the wards where other infected patients were hospitalized, whereas our WGS data suggested that this isolate was part of the same outbreak, clustering together with the rest of ST437 outbreak isolates based on core genome MLST. It was also confirmed that the same plasmid, carrying *bla*OXA-48, was present in the patient's isolate and other outbreak isolates showing the potential of WGS to enhance conventional epidemiologic investigation and allow a more accurate control of an outbreak [33].

We detected a combination of three different beta-lactamase genes *bla*OXA-48, *bla*NDM-1 and *bla*CTX-M-15 in six ST437 isolates (BR179-15, BR194-15, BR207-15, BR211-15, BR254-15, BR252-15), which to our knowledge is the first described combination of these genes in ST437. Co-existence of genes for CTX-M-15 and OXA-48 has been observed in *K. pneumoniae* isolate from our source patient (BR329-14) as well as in ST437 isolates in Serbia [12], where this patient was previously hospitalized [15]. We did not detect *bla*NDM-1 in *K. pneumoniae* isolates of the source patient, we did however detect it in *E. coli* and *Proteus mirabilis* isolate that were isolated from the same surveillance sample which was presumably the source for *bla*NDM-1 in *K. pneumoniae* outbreak isolates [15].

Five ST147 isolates, a clone associated with KPC, VIM, NDM, OXA-48-like and CTX-M-15 producers and a common ST in Europe, with prevalence in the Mediterranean basin, including North African countries [34,35], were detected. Four of them, closely related to each other according to WGS, were epidemiologically confirmed outbreak isolates and clustered together with only one allelic mismatch.

Moreover, four isolates of ST15, a pandemic clone [36], also detected in our neighbouring countries Croatia, Austria, and Hungary and associated with OXA-48, NDM-1 and CTX-M-15 producers [37–39], were detected in our selection, forming an unexpected outbreak. All four isolates shared the same beta-lactamase resistance pattern; however, additional epidemiological investigations have failed to reveal any clear connection.

Although it is one of the most commonly identified STs around the world and in Europe [4,40], ST258 was found in only one patient in our study which is to be expected as this ST is frequently associated with KPC carbapenemase, which is rare in Slovenia (less than 10% of CP-*K. pneumoniae*, Pirs M, personal communication). ST11, one of the most frequently detected STs in Europe [4,41], remained undetected among our isolates.

In the majority of European *K. pneumoniae* isolates that carried more than one carbapenemase gene, the *bla*OXA-48-like and *bla*NDM-like combination was reported to be the most common [4,7], and it is also the only combination detected in our *K. pneumoniae* isolates.

Although *bla*LEN was observed in ST258 isolate identified as *K. pneumoniae* according to short-read assembly, analysis of the hybrid assembly did not show the gene. However, *K. pneumoniae* belonging to ST258 and carrying *bla*LEN has already been described [42]. This discordance of results could be due to use of the two assemblers, which use different algorithms for assembly: Velvet assembler for the short-reads assembly and SPAdes for the hybrid-assembly [8,43].

It has been previously reported that outbreak isolates can have enhanced virulence potential [44], though our findings suggest that the virulence genes alone were not responsible for the successful spread of our outbreak clones, since they had a virulence score 0 and possessed a maximum of 11 virulence genes, whereas other isolates had a detected maximum of 26 genes, which is in accordance with some other studies [45,46].

A complete *mrk* cluster, coding type-3 fimbriae [22], was detected in majority of our isolates. We did not further investigate the missing genes in the *mrk* cluster in some isolates, although it has been reported that they could be interrupted by insertion sequences, which results in impaired detection [47]. A complete locus of an alternative siderophore-coding yersiniabactin was detected in few isolates, but none of them was an outbreak isolate. The yersiniabactin locus is often found in CP-*K. pneumoniae* and is strongly associated with isolates from the respiratory tract, including CP ST258 [48]. Our results are in concordance with this association, as all eleven genes of the locus were detected in our ST258 CP isolate. The aerobactin (*iuc*) locus, which consists of *iuc*A-D and *iut*A genes, has been known for its connection with virulence, and it was suggested its role in virulence is the most crucial among siderophore-coding loci [49]. We detected *iuc* locus in few of the outbreak isolates, which were without detected *ybt* locus. All of our outbreak isolates were therefore scored by virulence score 0, including majority of the remaining isolates, so our findings show that successful CP-*K. pneumoniae* clones present in Slovenia were not very virulent.

All of our isolates belonging to the largest cluster ST437 were assigned *wzi*-109, a combination that has been previously reported [50]. The *wzi* gene is a part of the *cps* locus responsible for synthesis of capsule polysaccharide and associated with virulence and capsular switching, important for escaping the host immune response [22,51]. The gene can thus be used for characterization and typing of *K. pneumoniae* isolates. Previous studies report a possible exchange of the *cps* locus, including *wzi* genes, between *K. pneumoniae* strains, although it is not clear if horizontal exchange and capsular switching is equally common in all clonal groups and STs [5,51]. Some correlation has been reported between certain *wzi* types and carbapenem-resistance genes, namely KPC-2, ST258 and *wzi*-29 [40,47]. *wzi*-29 was indeed detected in one of our isolates belonging to this well-known clone, which was positive for *bla*KPC-2. We detected two new *wzi* types.

IncL/M (pOXA-48), an epidemic plasmid connected with the worldwide dissemination of *bla*OXA-48 [29], was detected in our *bla*OXA-48 positive isolates, suggesting it could be responsible for the carbapenem resistance in *K. pneumoniae* isolates in Slovenian hospital. The genetic environment of *bla*OXA-48 was consistent with previous work [29]. Our findings regarding IncA/C plasmids could be compatible with Hancock et al. [52], highlighting an association with *bla*NDM, although this gene can also be associated with a number of other plasmids [53,54].

However, assembly of plasmids is difficult to achieve with short reads generated with Illumina Miseq [55], so further analysis is needed for determination of the plasmid responsible for acquired resistance determinants in our isolates. With long-read sequencing of five outbreak isolates, we confirmed almost exactly the same plasmid present in all five isolates (ST437 and ST147) positive for *bla*OXA-48, suggesting the plasmid-mediated spread of carbapenem-resistance among different STs and showing the importance of WGS in identifying and characterizing outbreaks.

## Conclusions

Whole-genome sequencing of a selection of Slovenian CR-*K. pneumoniae* isolates revealed multiple clusters of sequence types, of which two were involved in the single hospital outbreak of CP-*K. pneumoniae* in Slovenia. A further isolate belonging to the outbreak was identified and transmission of the *bla*OXA-48-carrying plasmid was confirmed, highlighting the importance of WGS in detecting and/or characterizing hospital outbreaks. A previously unidentified outbreak of ST15 isolates was unexpectedly discovered, a finding demonstrating the need for better surveillance of drug-resistant bacterial clones.

## Supporting information

**S1 Table. Quast and SeqSphere assembly quality report.**
(XLS)

**S2 Table. *K. pneumoniae* isolates (n = 32) characterization inlcuding resistance, plasmid and virulence profiles by WGS.**
(XLS)

**S1 File. Methods.** Six isolates were analysed using ONT GridIon. Four sequenced isolates were part of a confirmed outbreak: two belonged to the main cluster of ST437, the other two were selected as the only representatives of ST147 in order to confirm transmission of a plasmid with the *bla*OXA-48 resistance gene among different clones in the outbreak. One isolate of ST437 was previously classified as unrelated to the outbreak according to epidemiological data but clustered together with the outbreak isolates; therefore, we wanted to confirm it as part of the outbreak. The sixth isolate was sequenced because we detected *bla*LEN in *K. pneumoniae* belonging to ST258 and we wanted to confirm the presence of that gene.
(DOC)

**S1 Fig. EasyFig-generated schematic representation of *bla*OXA-48 encoding plasmids detected with long-read whole genome sequencing.** Black lines represent plasmids from five *K. pneumoniae* isolates positive for *bla*OXA-48 in PCR and short-read whole-genome sequencing. Coloured bars represent shared parts of genome between plasmids. An insertion was detected in isolate BR38 (white triangular insert in the bottom two coloured bars).
(TIF)

## Acknowledgments

We thank the team of curators of the Institut Pasteur MLST and whole-genome MLST databases for curating the data and making them publicly available at http://bigsdb.pasteur.fr/.

## Author Contributions

**Conceptualization:** Katarina Benulič, Mateja Pirš, Eva Ružić-Sabljić, Tjaša Cerar.

**Data curation:** Katarina Benulič, Mateja Pirš.

**Formal analysis:** Katarina Benulič, John W. A. Rossen, Tjaša Cerar.

**Investigation:** Katarina Benulič, Tjaša Cerar.

**Methodology:** Katarina Benulič, Mateja Pirš, John W. A. Rossen, Tjaša Cerar.

**Project administration:** Katarina Benulič.

**Resources:** Mateja Pirš.

**Software:** Katarina Benulič, Natacha Couto, Monika Chlebowicz, Tomaž Mark Zorec, Tjaša Cerar.

**Supervision:** Eva Ružić-Sabljić, Tjaša Cerar.

**Validation:** Katarina Benulič, Mateja Pirš, Tjaša Cerar.

**Visualization:** Katarina Benulič.

**Writing – original draft:** Katarina Benulič.

**Writing – review & editing:** Katarina Benulič, Mateja Pirš, Natacha Couto, Monika Chlebow-
icz, John W. A. Rossen, Tomaž Mark Zorec, Katja Seme, Mario Poljak, Tatjana Lejko
Zupanc, Eva Ružić-Sabljić, Tjaša Cerar.

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
