## [Decision Letter · Decision Letter 0]

8 Jan 2020

PONE-D-19-32138

Whole genome sequencing characterization of Slovenian carbapenem-resistant Klebsiella pneumoniae, including OXA-48 and NDM-1 producing outbreak isolates

PLOS ONE

Dear Dr. Benuli,

Thank you for submitting your manuscript to PLOS ONE. After careful consideration, we feel that it has merit but does not fully meet PLOS ONE’s publication criteria as it currently stands. Therefore, we invite you to submit a revised version of the manuscript that addresses the points raised during the review process.

We would appreciate receiving your revised manuscript by Feb 22 2020 11:59PM. To enhance the reproducibility of your results, we recommend that if applicable you deposit your laboratory protocols in protocols.io, where a protocol can be assigned its own identifier (DOI) such that it can be cited independently in the future. For instructions see: http://journals.plos.org/plosone/s/submission-guidelines#loc-laboratory-protocols

We look forward to receiving your revised manuscript.

Kind regards,

Rosa del Campo

Academic Editor

PLOS ONE

Journal Requirements:

2. We note that you are reporting an analysis of a microarray, next-generation sequencing, or deep sequencing data set. PLOS requires that authors comply with field-specific standards for preparation, recording, and deposition of data in repositories appropriate to their field. Please upload these data to a stable, public repository (such as ArrayExpress, Gene Expression Omnibus (GEO), DNA Data Bank of Japan (DDBJ), NCBI GenBank, NCBI Sequence Read Archive, or EMBL Nucleotide Sequence Database (ENA)). In your revised cover letter, please provide the relevant accession numbers that may be used to access these data. For a full list of recommended repositories, see http://journals.plos.org/plosone/s/data-availability#loc-omics or http://journals.plos.org/plosone/s/data-availability#loc-sequencing.

Reviewers' comments:

Reviewer's Responses to Questions

**Comments to the Author**

1. Is the manuscript technically sound, and do the data support the conclusions?

Reviewer #1: Partly

Reviewer #2: Yes

2. Has the statistical analysis been performed appropriately and rigorously? 

Reviewer #1: N/A

Reviewer #2: N/A

3. Have the authors made all data underlying the findings in their manuscript fully available?

Reviewer #1: Yes

Reviewer #2: Yes

4. Is the manuscript presented in an intelligible fashion and written in standard English?

Reviewer #1: Yes

Reviewer #2: Yes

5. Review Comments to the Author

Reviewer #1: In the present study, the authors describe the characterization of carbapenemase-producing Enterobacterales isolates recovered during two outbreaks in Slovenia (2014-2017) using whole genome sequencing. The introduction and Materials sections are well organized and presented, but the results section is very summarized and the discussion includes important results not previously mentioned in the results section. Moreover, a more rigorous analysis of plasmids must be done.

I have some major questions:

Line 120- Quality control and filtering of sequences have been performed?? Authors should make ensure that the sequences have enough quality to do the subsequent bioinformatics analysis. In the same way, they should include a quality assembly report (QUAST report or similar with N50, L50, number of contigs, length of genomes, etc.) that allows readers to know the quality of the draft genomes obtained after the assembly.

Line 208-210- A reference plasmid has not been represented along with plasmids identified during this study. To understand the epidemiological relevance of these OXA-48-encoding plasmids and their transmission in this country, they should be compared with the reference plasmids more closely related (for example the worldwide disseminated pOXA-48a described by Poirel in 2012). Alike, plasmid representation should be annotated (the genetic environment of carbapenemase genes plays an important role in the dissemination of these genes and should be represented in the figure 2).

Line 211- Plasmids with OXA-48 and NDM-1 were different? Information about this strain and both plasmids is not included in the study. Has the OXA-48-encoding plasmid been compared with the other 5 sequenced plasmids? These data must be included in the study.

Line 236- In the discussion, authors say that the combination blaOXA-48, blaNDM-1 and blaCTX-M-15 of genes is firstly detected in the ST437 clone. CTX-M enzymes has not been previously mentioned in the results section. If authors consider that this is an important finding, they should be included the result in the paragraph “Detection of beta-lactamase genes in WGS data” (not only in the table of supplementary material). The same in lines 238-240.

Line 262- Alike, blaLEN results have been firstly commented in the discussion and non-mention has been previously made in the Results.

Lines 295-301- These results should be better explained in the results section and then express a more general idea in the discussion.

Minor questions:

Line 58 and 224 - Authors comment that most frequent clones in Europe are ST11, ST15, ST101 and…ST285. With the latter the author must refer to clone ST258.

Line 59 - Authors should use “present simple” form throughout all the sentence (…ST101 are associated with carbapenemase….ST17 usually carry extended spectrum…)

Line 108 – Which are the breakpoints used for the antimicrobial susceptibility study? (EUCAST-2019 or 2018 or 2017??).

Lines 116-118 – Kits for library preparations are indicated in this section, but not the sequencing platform used (Illumina HiSeq? NovaSeq?)

Reviewer #2: This is a very interesting article presenting an outbreak caused by carbapenem resistant Klebsiella pneumoniae in Slovenia using WGS. Antimicrobial resistance markers, virulence factors, plasmid content and wzi types were determined with this teqnique which was not so accessible and possible in such a depth. This knowledge might be very helpful in future for possible surveillance and infectious control reasons especially for highly pathogenic bacteria such as K. pneumoniae.

6. PLOS authors have the option to publish the peer review history of their article (what does this mean?). If published, this will include your full peer review and any attached files.

Reviewer #1: No

Reviewer #2: No

---

## [Author Response · Author response to Decision Letter 0]

3 Mar 2020

Line 120- Quality control and filtering of sequences have been performed?? Authors should make ensure that the sequences have enough quality to do the subsequent bioinformatics analysis. In the same way, they should include a quality assembly report (QUAST report or similar with N50, L50, number of contigs, length of genomes, etc.) that allows readers to know the quality of the draft genomes obtained after the assembly.

We have included the information in the revised version of the article. Please see lines 121-127 and S1 Table.

Line 208-210- A reference plasmid has not been represented along with plasmids identified during this study. To understand the epidemiological relevance of these OXA-48-encoding plasmids and their transmission in this country, they should be compared with the reference plasmids more closely related (for example the worldwide disseminated pOXA-48a described by Poirel in 2012). Alike, plasmid representation should be annotated (the genetic environment of carbapenemase genes plays an important role in the dissemination of these genes and should be represented in the figure 2).

Our five ONT sequenced pOXA-48 positive isolates are now annotated and compared to the reference plasmid described by Poirel et al. (2012). Please see lines 149-155, 224-227, line 321 and Figure 2.

Line 211- Plasmids with OXA-48 and NDM-1 were different? Information about this strain and both plasmids is not included in the study. Has the OXA-48-encoding plasmid been compared with the other 5 sequenced plasmids? These data must be included in the study.

Thank you for your comment. In the strain with both, OXA-48 and NDM-1 plasmid (BR252-15), OXA-48 plasmid was compared to four other sequenced plasmids and the comparison is seen in the Figure 2. In the 6th ONT sequenced isolate, as expected, we did not detect OXA-48 or NDM-1 plasmid. Please see lines 102-104 and also lines 229-232.

Line 236- In the discussion, authors say that the combination blaOXA-48, blaNDM-1 and blaCTX-M-15 of genes is firstly detected in the ST437 clone. CTX-M enzymes has not been previously mentioned in the results section. If authors consider that this is an important finding, they should be included the result in the paragraph “Detection of beta-lactamase genes in WGS data” (not only in the table of supplementary material). The same in lines 238-240.

Thank you for your comment. The information was added, please see lines 178-180.

Line 262- Alike, blaLEN results have been firstly commented in the discussion and non-mention has been previously made in the Results.

The information was added, please see lines 178-180.

Lines 295-301- These results should be better explained in the results section and then express a more general idea in the discussion.

Changes were made to the results and discussion sections regarding the plasmids. Please see lines 220-222 and also lines 317-324.

Line 58 and 224 - Authors comment that most frequent clones in Europe are ST11, ST15, ST101 and…ST285. With the latter the author must refer to clone ST258. 

The mistake was corrected.

Line 59 - Authors should use “present simple” form throughout all the sentence (…ST101 are associated with carbapenemase….ST17 usually carry extended spectrum…)

The sentence was corrected.

Line 108 – Which are the breakpoints used for the antimicrobial susceptibility study? (EUCAST-2019 or 2018 or 2017??).

For AST with disk diffusion contemporary EUCAST guidelines were used (with the ending of the isolate name denoting the year of isolation). The information was added to lines 109-110.

Lines 116-118 – Kits for library preparations are indicated in this section, but not the sequencing platform used (Illumina HiSeq? NovaSeq?)

The information regarding sequencing platform was added. Please see lines 120-121.

Please note: There was a minor change to S1_File_Methods. Figure S2, which was not referred to in the manuscript and not intended for publication, was accidentally included in the previous submitted version. It has been removed from the current submission.

---

## [Editor Report · Decision Letter 1]

25 Mar 2020

Whole genome sequencing characterization of Slovenian carbapenem-resistant Klebsiella pneumoniae, including OXA-48 and NDM-1 producing outbreak isolates

PONE-D-19-32138R1

Dear Dr. Benulic,

We are pleased to inform you that your manuscript has been judged scientifically suitable for publication and will be formally accepted for publication once it complies with all outstanding technical requirements.

With kind regards,

Rosa del Campo

Academic Editor

PLOS ONE
---

## [Editor Report · Acceptance letter]

26 Mar 2020

PONE-D-19-32138R1 

Whole genome sequencing characterization of Slovenian carbapenem-resistant *Klebsiella pneumoniae*, including OXA-48 and NDM-1 producing outbreak isolates 

Dear Dr. Benulic:

I am pleased to inform you that your manuscript has been deemed suitable for publication in PLOS ONE. Congratulations! Your manuscript is now with our production department. 

With kind regards,

on behalf of

Dr. Rosa del Campo 

Academic Editor

PLOS ONE